# A Comparison of Sedentary Behavior as Measured by the Fitbit and ActivPAL in College Students

**DOI:** 10.3390/ijerph18083914

**Published:** 2021-04-08

**Authors:** Chelsea Carpenter, Chih-Hsiang Yang, Delia West

**Affiliations:** 1Department of Clinical and Health Psychology, College of Public Health and Health Professions, University of Florida, Gainesville, FL 32610, USA; clarsen1@phhp.ufl.edu; 2Department of Exercise Science, Arnold School of Public Health, University of South Carolina, Columbia, SC 29208, USA; cy11@mailbox.sc.edu

**Keywords:** sedentary behavior, measurement, Fitbit

## Abstract

Previous studies have examined the ability of the Fitbit to measure physical activity compared to research-grade accelerometers. However, few have examined whether Fitbits accurately measure sedentary behavior. This study examined whether the Fitbit Charge 3 adequately quantifies sedentary behavior compared to the gold standard in objectively measured sedentary behavior assessment, the activPAL. Eleven adults wore a Fitbit Charge 3 and activPAL device for 14 days and self-reported their sedentary behavior each week. ActivPAL epoch data were summed into minute-by-minute data and processed with two cutpoints (*activPAL_Half* and *activPAL_Full*) to compare to Fitbit data. Paired t-tests were used to examine differences between the two devices for sedentary behavior variables. Intraclass correlations were used to examine device agreement. There was no significant difference in sedentary time between *activPAL_Half* and Fitbit data, but *activPAL_Full* estimated significantly lower sedentary time than Fitbit. Intraclass correlations showed high agreement. We suggest that Fitbit could replace activPAL when measuring total sedentary time.

## 1. Introduction

Emerging evidence indicates that excessive sedentary time is associated with a multitude of poor health outcomes, including an increased risk of cardiovascular disease [1,2], type 2 diabetes [3], and some cancers [4]. Data also suggest the pattern in which sedentary time is accumulated matters, with prolonged bouts of sedentary time associated with adverse cardiometabolic outcomes [5,6]. This detrimental relationship between sedentary behavior and health has compelled a focus on the accurate measurement of sedentary behavior to help further elucidate its effects.

Sedentary behavior can be quantified with subjective and objective measures, and the inclusion of both has been strongly recommended in studies examining the influence of sedentary behavior on health [7]. Research-grade accelerometers and inclinometers that objectively assess sedentary behavior offer greater accuracy than self-report measures [7]. The activPAL (Physical Activity Technologies, Glasgow, Scotland) is a popular research-grade inclinometer designed to measure free-living activity and classify it as sitting, standing, and stepping. The activPAL can distinguish between sitting and standing, which is essential for the accurate classification of sedentary time [8], and this unique feature of the device makes it the “gold standard” for objective sedentary behavior measurement [9,10,11].

Consumer-grade accelerometers (i.e., wearables) can also measure activity in free-living settings. Popular wearable devices such as the Fitbit (Fitbit, San Francisco, CA, USA) can be worn continuously on the wrist, and in addition to the number of steps and minutes of physical activity, provide feedback on sedentary behavior. There are some indications that both wrist- and waist-worn Fitbits can capture moderate-to-vigorous physical activity levels with reasonable accuracy when compared to research-grade accelerometers [12,13], although there are also some suggestions that both types of Fitbit devices may overestimate active time [14,15]. Given the popularity, ease of use, and relatively low cost of devices like Fitbits, they present an attractive alternative to research-grade devices for research on sedentary behavior.

Little is known, however, about the performance of Fitbit devices for measuring sedentary behavior. The only study to examine the Fitbit for measuring sedentary behavior relative to the gold-standard activPAL was limited in that it compared the two devices for only a single day and examined a waist-worn Fitbit device (Fitbit One) [16]. To our knowledge, there are no data currently available to determine the agreement of data from a wrist-worn Fitbit tracker with state-of-the-art objective measurement of sedentary behavior using activPAL in a free-living environment over multiple days to determine sedentary patterns.

Thus, the purpose of this study was to determine whether the Fitbit Charge 3 (from here on, simply Fitbit) provides comparable estimates of time spent sedentary compared to the activPAL, whether the two devices offer similar classification of an individual as sedentary in free-living conditions, as well as compare participants’ compliance with wearing each device. In addition, the study examines different approaches to classifying sedentary behavior using activPAL epoch data. The activPAL software provides both event-based and epoch (time-based) data. The event-based data are typically used for classifying individuals as sedentary and there are no guidelines that establish how much of a minute in the epoch-based data must be spent engaged in sedentary behavior to classify that minute as sedentary. Therefore, we explored two different thresholds (half minute and full minute) to determine which classification scheme agrees most closely with the events data. This comparison advances conversations about how sedentary behavior might best be classified using this gold-standard device and also provides two epoch-based metrics from the activPAL with which to compare Fitbit minute-by-minute data.

## 2. Materials and Methods

### 2.1. Study Overview

Young adults were recruited via emails detailing study information and by referral from other study participants. Eligibility criteria included being (1) 18 years or older, (2) a current undergraduate student at the University of South Carolina, and (3) a smartphone owner. There were no eligibility criteria about sedentary time, and sedentary behavior was mentioned in the title of the study only. If interested and eligible, individuals were given a Fitbit device and loaned an activPAL to wear simultaneously for two weeks. After a brief orientation to both devices, participants were provided with contact information in the event they encountered difficulties and were scheduled to return after two weeks. Participants were paid up to $100 for completing questionnaires and returning the activPAL at the end of the second week. The study was conducted in accordance with the Declaration of Helsinki, and the protocol was approved by the University of South Carolina Institutional Review Board (Pro00090225, 7 December 2019). All participants gave their informed consent before inclusion in the study.

### 2.2. Device-Measured Daily Movement Behaviors

#### 2.2.1. Fitbit

Participants wore a Fitbit tracker (Fitbit Charge 3) on their non-dominant wrist. The Fitbit (Charge 3) is a triaxial accelerometer that tracks multiple measures of physical activity (i.e., steps, stairs climbed, active minutes) and heart rate. These data are then classified as sedentary, lightly active, fairly active, or very active using Fitbit proprietary algorithms. Validation studies have shown wrist-worn Fitbit models, such as the Charge 3, to accurately assess the steps of adults in both lab-based [17,18,19] and free-living [15,20] conditions.

Participants were instructed to wear the Fitbit, even while sleeping, for the full two-week study duration. To ensure continuous data collection, the “all day sync” feature was selected to enable the participant’s Fitbit to sync wirelessly to the smartphone application via Bluetooth at random points throughout the day. Data were transferred from Fitbit, Inc. through an application programming interface (API) that downloaded all physical activity, sedentary behavior, and sleep time data every 10 min to the study website, allowing real-time data capture. Each Fitbit was registered to the study with a unique identifier (rather than to the individual) to minimize privacy concerns. No personal information was attached to the Fitbit device. Variables derived from Fitbit, Inc. and considered in the current analyses included heart rate, sleep time, and minute-by-minute activity data, including minutes classified as sedentary.

#### 2.2.2. ActivPAL

The activPAL4 (Pal Technologies Ltd., Glasgow, UK) is a thigh-worn inclinometer shown to be a valid and reliable measure of sedentary behavior [9,11]. The device was waterproofed and adhered to the upper right thigh with non-allergenic adhesive tape. Participants were instructed to wear the device continuously for 24 h for 14 days, even while showering or participating in water-based activities. In addition, participants completed a sleep log and were asked to record the time and reason if the device was removed. Time spent sitting/lying, standing, and stepping were calculated using activPAL’s proprietary software. Data were exported from activPAL in the events format and 15 s epochs format, with every 4 epochs being summed and constituting each minute.

#### 2.2.3. Sleep Time

Sleep time is not considered sedentary time [8], so it was removed to determine the waking day and to allow the proportion of the day spent sedentary to be calculated. Sleep time was determined using both the participant’s self-reported sleep log and Fitbit’s objectively measured sleep log. A conservative approach was used to determine sleep time such that if a minute fell in the window of either the self-reported sleep log or the Fitbit sleep log, it was classified as sleep time. In addition, any naps recorded by the participant or by the Fitbit were classified as sleep time.

#### 2.2.4. Wear Time

A valid day of data for the activPAL was defined as at least 10 h of waking wear time [21]. Non-wear time was defined as intervals of at least 90 consecutive minutes with 0 counts per minute (cpm), allowing for 2 min of observations with counts <100 cpm within this period. A similar 10 h daily wear-time criterion was used for Fitbit to consider the Fitbit data valid; if heart rate data were available for the minute in question, it was assumed that the participant was wearing the device.

#### 2.2.5. Quantification of Sedentary Time

To allow the examination of the agreement between the sedentary quantification by the two devices, we used the 15 s epoch datafile and summed every 4 epochs into minute-by-minute data to provide comparable data to the Fitbit metrics, which only provides data aggregated to the minute level. Without a current guideline on the best approach for classifying a given minute as sedentary, we classified sedentary time using two different criteria: (1) if over half of the minute (>30 s) was spent sitting/lying down (*activPAL_Half*), and (2) if the entire 60 s were spent sedentary (*activPAL_Full*). Fitbit uses a proprietary algorithm, so we kept its default classification of sedentary time. Both the *activPAL_Half* and *activPAL_Full* approaches were used to compare with Fitbit estimates of time spent sedentary to determine their relative congruency. These analyses were conducted using temporally aligned minute-by-minute valid wear-time data from both devices.

This study also compared the estimated sedentary time derived from the activPAL event datafile and the epoch datafile to determine which activPAL epoch classification method (*activPAL_Half* or *activPAL_Full*) yielded estimates of sedentary time similar to those calculated from the event data. The two datafiles present different information and different perspectives on how to classify sedentary behavior. In short, events data provide information on bouts of sedentary behavior (number and duration) and epoch data offer estimates of time engaged in sedentary behavior. Specifically, the events datafile provides a list of all of the bouts of sitting/lying, standing, and stepping, with the time each bout begins and ends. This file provides more precise data for each activity, as well as information about the activity bout frequency and duration. The 15 s epoch datafile, on the other hand, reports the number of seconds in each posture, the number of steps, and the number of sit-to-stand transitions during that window, irrespective of bout and with no indication of the order in which each activity occurs [22]. In addition, it should be noted that the activPAL’s epoch and event data can yield different estimates of waking day hours. Most studies use a log on which participants are asked to self-report sleeping time and any non-wear time. This information can be used to parse out waking wear time. For the 15 s epoch datafile, self-reported waking wear time is simply matched to the file. For the events datafile, however, self-report waking wear-time periods are matched with “bouts” of activity and therefore may lead to a discrepancy between the two datafiles in the amount of waking wear time [22]. Even though this can affect estimates of total minutes spent sedentary, it does not impact calculations of the proportion of the day spent sedentary. Therefore, we focused on the percentage of the day spent sedentary to compare the approaches.

### 2.3. Classification of Individuals

Using previously established criteria, participants were considered to be sedentary if they engaged in an average of 7 h of sedentary behavior per day over the 2 weeks [23,24]. Individuals were classified as sedentary based on the Fitbit and *activPAL_Events* measures and these classifications were compared.

### 2.4. Self-Reported Sedentary Behavior

Self-reported total sedentary time was assessed two times (end of week 1 and end of week 2) using the short version of the International Physical Activity Questionnaire (IPAQ) [25] and the Sedentary Behavior Questionnaire (SBQ) [26]. The IPAQ queries about sedentary minutes using the question, “During the last seven days, how much time did you usually spend sitting on a week day?” It is considered an acceptable reliable and valid measure of sedentary time [27]. The SBQ assesses the time an individual reports engaging in eight typical sedentary activities over the previous week. Sedentary time is measured for both weekdays and weekend days with the question “On a typical week (end) day, how much time do you spend (from when you wake up until you go to bed) doing the following … ” and by listing the sedentary activities [26]. Response options range from none to 6 or more hours. Average daily sedentary time was calculated using a weighted average of weekday and weekend day. Responses were truncated to 1440 for values greater than 1440 min (i.e., 24 h) [26,28].

### 2.5. Anthropometrics and Sociodemographics

Height and weight were measured to the nearest centimeter and 0.1 kg, respectively, at baseline. Body mass index (BMI: kilograms/meters^2^) was calculated. In addition, age, sex, race/ethnicity, college major, and current class standing were self-reported at baseline.

### 2.6. Cost

The costs of the Fitbit and activPAL devices were recorded by investigators.

### 2.7. Statistical Analyses

Descriptive statistics were used to determine the number of individuals for whom the sedentary classification by the Fitbit and activPAL events data were the same or different. The *activPAL_Events* estimate was used to make the comparison because it is the “gold standard” approach used when classifying sedentary behavior in other studies using the activPAL. We elected not to examine the correspondence between sedentary classifications from the Fitbit and all three activPAL measurements because the other metrics are not the standard in the field for determining sedentary classification. Paired *t*-tests were used to compare the average daily hours of sedentary behavior and percentage of the waking day spent sedentary between the Fitbit estimates and the three activPAL estimates (half, full, and event), as well as between the pairs of the activPAL estimates. Intraclass correlation coefficients (ICCs) were used to make paired comparisons of the degrees of correlation and agreement between the Fitbit and each of the epoch-based activPAL estimates (*activPAL_Half*, *activPAL_Full*). The ICCs were interpreted as agreement using standard benchmarks: 0.00 to 0.20 slight, 0.21 to 0.40 fair, 0.41 to 0.60 moderate, 0.61 to 0.80 substantial, and 0.81 to 1.00 almost perfect [29]. Pearson correlations were used to examine whether the self-report and objective measures of sedentary time were correlated.

## 3. Results

The participants (*n* = 11) were predominately white (91%) females (73%) with an average BMI of 23.5 ± 3.9 kg/m^2^ and an average age of 20.7 ± 0.5 years. Compliance with the activPAL protocol was modestly but significantly higher than observed with the Fitbit during the two-week period, with an average daily wear time of 23.7 ± 0.7 (range: 21.6–24) hours per day for the activPAL and an average daily wear time of 20.2 ± 1.9 (range: 17.6–23.3) hours per day for the Fitbit (*p* < 0.001).

The average waking time the participants wore both devices concurrently was 12.2 ± 4.5 h per day. When examined by activPAL_Epoch, Fitbit, and *activPAL_Events*, waking time was 15.9 ± 0.8 h, 14.8 ± 0.8 h, and 15.2 ± 1.0 h, respectively. Participants wore both devices for at least 10 h (a valid day) for an average of 11 days (range: 8–14), with an average waking time of 14.5 ± 1.7 h per valid day.

The average daily sedentary behavior, in both hours and percent of the waking day, is described in Table 1. The Fitbit and “gold standard” *activPAL_Events* estimates were not significantly different from one another (*p* > 0.05) for either waking hours spent sedentary or percentage of the day spent sedentary. Using the criterion of an average of 7 h/day to classify individuals as sedentary [23,24], there was 91% agreement in sedentary classification between the Fitbit and *activPAL_Events* (i.e., 10 out of 11 were classified similarly). Analyses examining which epoch metric (*activPAL_Half* or *activPAL_Full*) most closely resembled estimates derived from the “gold standard” *activPAL_Events* data showed that *activPAL_Full* yielded significantly lower estimates of sedentary time (*p* < 0.0001) than observed with the event-based data, but there were no significant differences between *activPAL_Half* and *activPAL_Events* estimates. Therefore, the *activPAL_Half* classification was used in ICC analyses to compare the minute-by-minute data from the activPAL and Fitbit. The sedentary behavior time estimates derived from the Fitbit were very similar to those derived from the activPAL, with an ICC of 0.942 (95% CI: 0.922–0.958) between the Fitbit and *activPAL_Half*.

Self-reported sedentary time was 9.0 ± 4.4 h and 6.1 ± 2.7 h for the SBQ and the IPAQ, respectively. However, neither IPAQ nor SBQ self-reported sedentary time were significantly correlated with Fitbit or any of the activPAL objectively measured sedentary time estimates (Table 2).

The cost of the Fitbit Charge 3 at the time of this study was USD 99. The activPALs were obtained at a price of USD 348 per device.

## 4. Discussion

These data suggest a Fitbit Charge 3 device may offer a reasonable approach to measuring sedentary behavior in research settings. Estimates of the average total number of sedentary minutes in a day from the Fitbit were comparable to those derived from the gold-standard activPAL device. Both devices estimated that college students spent over 9 h engaged in sedentary behavior, which corresponds to data from a recent systematic review and meta-analysis [30]. Furthermore, the Fitbit was comparable to activPAL for classifying an individual as sedentary, with 91% agreement between the classifications made by each device. However, there was no correlation between the two self-report measures of sedentary time and any of the objective measures of sedentary time in this study. Lastly, adherence to the wear-time protocol was high for each device, with an average wear time of over 20 h per day for both devices for the 14-day study period.

Previous studies examining Fitbits relative to research-grade accelerometers and inclinometers have had conflicting results. Two studies suggested that wrist- and waist- worn Fitbits underestimate sedentary time relative to a non-inclinometer accelerometer [15,20], whereas another indicated sedentary time captured with a wrist-worn Fitbit device is comparable to assessments with a non-inclinometer accelerometer [14]. Non-inclinometer accelerometers are themselves problematic for accurately measuring sedentary behavior. The mixed findings may be due to their inability to take posture into account, thereby missing a critical element of the operational definition of sedentary behavior [8]. Only one study to date has compared a Fitbit to the gold standard for assessing sedentary behavior, the activPAL. Those results suggest that the waist-worn Fitbit underestimated sedentary behavior relative to the activPAL [16], which conflicts the findings in this study. The discrepancy could be attributed to differences in Fitbit device placement (i.e., waist- vs. wrist-worn device), which has been shown to be an influential factor in measurement variability [31]. In addition, the current study had a longer study duration than the previous study (i.e., 14 days vs. 1 day) and a different study population (college students vs. middle-aged adults). The minimal differences in sedentary behavior we observed suggest that Fitbits may be better suited for the extended characterization of habitual sedentary patterns rather than examining correspondence between devices for a single day.

Although activPAL data can be exported in epochs or events, using the events file is the recommended approach [22]. However, if the research question entails categorizing a minute as sedentary or non-sedentary, the epoch data might be better suited to answer the question [10,32]. Our methodological evaluation considered both data formats and provides insight into the agreement between epoch and events data. The half-minute epoch data yielded an estimate of sedentary time more similar to the events data than the full-minute approach did, which makes sense, as the full-minute approach is more conservative with a higher threshold for classifying a minute as sedentary. When comparing Fitbit data to the half- and full-minute approaches, Fitbit data overestimated sedentary time compared to the *activPAL_Full* approach but were highly consistent with and not significantly different from the *activPAL_Half* approach, underscoring the likelihood that the Fitbit device would be appropriate in studies seeking to quantify minutes of sedentary behavior.

Participants wore each device for an average of over 20 h per day during the 14-day study. The similar wear time is noteworthy since the activPAL is waterproof, adhered to the thigh, and is only removed for specific circumstances (e.g., going to the ocean, irritation) for the full study duration. On the other hand, the Fitbit is often removed for water-based activities and requires charging every 5–7 days. The high concurrence in wear times indicates that participants actively chose to continuously wear the Fitbit device during the study period. The high compliance seen in this sample could be attributed to the 24 h wear-time protocol [33] and/or the high acceptability of each device. Even though this was likely a highly compliant sample consisting of motivated individuals getting incentives, the compliance in this study mirrors data from seven-day studies that used the ActiGraph (ActiGraph, Pensacola, FL), which show that participants wore the device for 5.5–5.8 days [14,34] and had 14.9 h of waking time [14]. The comparable compliance for the two devices underscores the potential of the Fitbit to be used for sedentary behavior measurement and provides confidence in the Fitbit as an alternative to activPAL when needed for the quantification of sedentary time.

There was no observed significant correlation between the subjective and objective measures of sedentary behavior. Discrepancies between the two measurement types have been noted previously in the literature, with a recent systematic review noting that self-report measures tend to have large bias and poorer precision than objective measures [35]. However, recommendations are to include both since each communicates different aspects of sedentary behavior [7], making self-report measures a poor substitute for objective measures, and vice versa. Therefore, even with an objective measure such as the Fitbit, it may be advisable to include a self-report measure of sedentary time.

The study has several strengths worth noting. Most studies comparing the Fitbit to a research-grade accelerometer have used the ActiGraph, which is not considered the gold standard for sedentary behavior research [9]. In addition, the 14-day study duration provided an extended and stable characterization of habitual sedentary behavior and accounted for both weekday and weekend day sedentary patterns. Compliance to the study protocol was high with an average of 11 days or 79% of the study period having valid data. However, there are also several limitations in this study, including a small homogenous adult sample that included predominately white college students. Studies with other populations, such as children or older adults, may not yield the same results given the differing activity patterns of these populations. In addition, the consistency in sedentary behavior assessment with the Fitbit and activPAL observed in this study may not signify consistency between the two devices for other physical activities, which give these data no bearing on the full spectrum of physical activity.

## 5. Conclusions

The Fitbit was comparable in its measurement of sedentary time to the gold-standard activPAL measure. Considering Fitbits cost a fraction of the price of the activPAL and yield similar sedentary estimates and comparable compliance data to the activPAL, the Fitbit offers a reasonable alternative to the activPAL and may be a viable option for large, budget-conscious studies interested in the measurement of sedentary behavior.

## Figures and Tables

**Table 1 ijerph-18-03914-t001:** Sedentary time and percent sedentary waking time on valid days for both the activPAL and Fitbit device.

ID	Number of Valid Days	Average Waking Time on Valid Days	activPAL_Full	activPAL_Half	activPAL_Events	Fitbit	Agreement	activPAL_Full	activPAL_Half	activPAL_Events	Fitbit
1	8	14.6 ± 2.1	8.3 ± 2.3	9.0 ± 2.3	9.6 ± 2.2	9.7 ± 2.3	a	56.5	61.3	62.8	66.4
2	12	13.4 ± 1.4	6.1 ± 2.3	6.7 ± 2.2	6.6 ± 2.6	7.5 ± 1.7	b	47.3	51.3	50.0	58.0
3	13	15.3 ± 2.1	10.9 ± 1.5	11.5 ± 1.5	11.8 ± 1.8	11.7 ± 1.5	a	71.3	74.9	75.0	76.6
4	12	14.9 ± 1.6	6.7 ± 1.8	7.3 ± 1.8	7.4 ± 1.8	8.9 ± 1.7	a	44.9	49.2	49.9	59.7
5	12	14.5 ± 1.9	8.5 ± 1.7	9.3 ± 1.8	10.3 ± 1.8	9.6 ± 1.7	a	59.1	64.5	64.7	66.1
6	13	15.1 ± 1.4	7.4 ± 1.7	8.4 ± 1.8	9.9 ± 1.8	8.3 ± 1.7	a	49.2	55.5	59.9	55.3
7	11	13.5 ± 2.1	5.4 ± 1.9	6.1 ± 1.9	6.7 ± 2.6	6.5 ± 1.6	a	40.1	45.2	46.2	48.3
8	8	15.6 ± 1.8	10.5 ± 3.6	11.2 ± 3.8	11.7 ± 2.5	11.1 ± 3.3	a	67.6	72.1	71.8	71.4
9	9	13.2 ± 1.6	8.5 ± 2.1	9.0 ± 2.2	9.3 ± 2.6	9.8 ± 2.2	a	64.3	68.7	68.0	78.4
10	9	15.0 ± 1.2	8.6 ± 2.3	9.3 ± 2.3	9.7 ± 1.9	8.7 ± 2.0	a	57.0	62.2	62.1	58.1
11	14	14.8 ± 1.8	9.5 ± 2.0	10.2 ± 2.1	10.8 ± 2.3	9.3 ± 2.1	a	63.8	68.4	67.1	62.7
All	11	14.5 ± 1.7	8.2 ± 2.1	8.9 ± 2.2	9.4 ± 2.2	9.2 ± 2.0		56.5	61.2	61.6	63.4

(a) Agreement between Fitbit and activPAL_Events on sedentary classification. (b) Disagreement between Fitbit and activPAL_Events on sedentary classification. activPAL_Half: A minute was classified as sedentary if over half of the minute (>30 s) was spent sitting/lying down. activPAL_Full: A minute was classified as sedentary if the full 60 s were spent sitting/lying down.

**Table 2 ijerph-18-03914-t002:** Correlation between minutes of self-reported and objectively measured sedentary behavior.

	activPAL_Full	activPAL_ Half	activPAL_Events	Fitbit
IPAQ	−0.272	−0.248	−0.443	−0.154
SBQ	−0.097	0.018	−0.055	−0.176

IPAQ = International Physical Activity Questionnaire. SBQ = Sedentary Behavior Questionnaire.

## Data Availability

The data presented in this study are available on request to qualified researchers from the corresponding author.

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
