# Peer review of "A Comparison of Sedentary Behavior as Measured by the Fitbit and ActivPAL in College Students"

_ijerph, 2021, doi:10.3390/ijerph18083914_

Round 1
Reviewer 1 Report
The current study compares Fitbit charge 3 and ActivPAL measures of sedentary behavior, with the latter considered as the gold standard. The study indicates a high level of agreement between the two devices. The strengths of the study are the usage of the gold standard device for sedentary behavior and the comparison of different metrics (time and classification) and instruments (gold standard and questionnaires). The limitations of the study are insufficient background, statistical analysis, and the presentation of the results.
Abstract:
The authors use Fitbit in a generic way, although they examined only a specific model: Fitbit Charge 3. Please, specify the model or state: "from here on simply Fitbit".
Line 21
is the word "if" correct or do you mean "in"?
Introduction
According to the authors, previous studies examined the validity of Fitbit devices on measuring physical activity but very few examined it on measuring sedentary behavior. However, Ferguson et al. and Brewer et al. compared various models, while Redenians et al. and Reid et al, compared Fitflex and One and just Fitbit flex, respectively, with research-grade accelerometers. Thus, it is important in the introduction to deepen this part and to define the specific model adopted in this study.
Material and methods
line 63: Please, provide here participants' characteristics.
line 81-83: The authors refer here to a series of studies using different Fitbit models, including those devices that are worn on the waist. Please, provide references for the investigated model if available, or justify why citing studies in which different models were used. This aspect can be crucial and must be discussed.
Line 124: "we used the 15-second epoch approach and summed every 4 epochs into minute-by-minute data". Please, put here a reference (e.g., Bassett et al. 2013);
Line 126 - 134: considering the lack of guidelines on the best approach for defining a given minute as sedentary, the authors compared two different methods for classifying sedentary behavior. This is an original aspect investigated in the present study and should be presented in the introduction as an objective of the study and not in the methods section.
Line 172: Statistical analysis is confusing. Paired sample t-test seems inadequate to compare the variables investigated because there are more than two averages.
Line 175: Did the authors run the Bland Altman test for investigating the agreement? Why the authors chose the ICC? This part is not clear also because the authors used the terms agreement and correspondence interchangeably. When an agreement is considered discordant and when concordant?
Results
Table 1 is not well-formatted and I cannot read the right side.
Discussion
Please, when you refer to previous studies, specify the specific model investigated.
Reviewer 2 Report
Interesting and valid-investigative project that will benefit other researchers attempting to categorize sedentary time at a low cost using an accelerometer (vs using an expensive inclinometer).
Your research design and use of the Fitbit and activePAL activity trackers were very straightforward. I commend you for tackling "over the counter" technology in the form of the Fitbit vs the research-grade activePAL. Your classifications of activPAL_Half and activPAL_Full were appropriate and made a good case for either category aligning with Fitbit, which samples over a minute. Lastly, it's tough to know what's inside the "black box" of Fitbit's classification of sedentary time; thus, I agree with you using the "default classification" for sedentary time when using the cheaper device. It's a real world application of the tracker.
I really only have one concern or need for clarification:
Page 3 - line 136+: It's not clear what you mean by "event data". You also go on in this paragraph to say "activePAL's epoch and event data have different estimates of waking day hours" - explicitly explain this in greater detail or why this difference exists, including a complete or holistic overview of what "event data" is measuring and/or how it's measured or calculated. This seems important to tease out. However, I understand you're more concerned with measuring sedentary time. Still, thanks for the clarification.
Lastly, have you considered reporting heart rate from the Fitbit as a way to give "objective" context to waking hours and/or sedentary time? I don't think the activePAL records HR; therefore reporting HR via the Fitbit would be more of an added descriptive variable. Thoughts? I think it might enhance your Table 1, for instance.
**Please review my attachment of your manuscript with YELLO HIGHLIGHTS and comments associated with each highlighted area. There were a few typos and wording issues I wanted to draw your attention to. Thanks.

Reviewer 3 Report
Authors compared two devices for activity monitoring, a commercial one, and one used for research purposes. Particularly, the novelty of the proposed paper, is to use these devices to measure sedentary activity.
In vivo experiments have involved 11 subjects, and they have shown no significant differences between the two devices.
The article is technically sound and experiments have been correctly conducted.
I suggest for future work to include a higher number of subjects, and study if significant differences can be observed among different age ranges. Furthermore I suggest to repeat this study with cheaper devices, such as Xiaomi band.
Round 2
Reviewer 1 Report
The authors did good work in editing the manuscript. However, I still have major concerns relative to the statistical approach and results presentation. My comments are listed below point-by-point:
- The authors use Fitbit in a generic way, although they examined only a specific model: Fitbit Charge 3. Please specify the model or state “from here on simply Fitbit”
- To clarify the model used in this study, we changed the abstract to say “This study examined whether the Fitbit Charge 3 adequately quantifies sedentary behavior when compared to the activPAL” (line 12) and also changed the study purpose to say “The purpose of this study was to determine whether the Fitbit Charge 3 (from here on, simply Fitbit) provides comparable estimates…” (line 61-62).
- I am satisfied with the changes made by the authors.
- Line 21- Is the word “if” correct or do you mean “in”?
- We have revised the sentence on line 21.
- I am satisfied with the changes made by the authors.
- According to the authors, previous studies examined the validity of Fitbit devices on measuring physical activity but very few examined it on measuring sedentary behavior. However, Ferguson et al and Brewer at al compared various models, while Redenians et al and Reid compared Fitbit Flex and One and just Fitbit Flex, respectively, with research grade accelerometers. Thus, it is important in the introduction to deepen this part and to define the specific model adopted in this study.
- The authors have revised the introduction to reflect the types of models (wrist- vs waist-worn) that were used in the validation studies, and to cite the noted studies. However, we want to underscore that these articles examined agreement of sedentary behavior using accelerometers, as the reviewer notes, and these devices are considered the gold standard for measuring physical activity but not the gold standard for measuring sedentary behavior. Thus, we believe that our manuscript fills an important gap in the literature in that we compare Fitbit measures with those from activPAL devices, which represent the current gold standard approach to measuring sedentary behavior. The revised sentences and their line numbers are included below.
- 46-52: There are some indications that both wrist and waist worn Fitbits can capture moderate-to-vigorous physical activity levels with reasonable accuracy when compared to research-grade accelerometers [12,13], although there are also some suggestions that the both types of Fitbit devices may overestimate active time [14,15]. Given the popularity, ease of use, and relatively low cost of devices like Fitbits, they present an attractive alternative to research-grade devices for research on sedentary behavior.
- 53-60: Little is known, however, about the performance of Fitbit devices for measuring sedentary behavior. The only study to examine the Fitbit for measuring sedentary behavior relative to the gold standard activPAL was limited in that it compared the two devices for only a single day and examined a waist-worn Fitbit device (Fitbit One) [16]. To our knowledge, there are no data currently available to determine the agreement of data from a wrist-worn Fitbit tracker with state-of-the-art objective measurement of sedentary behavior using activPAL in a free-living environment over multiple days to determine sedentary patterns.
- I am satisfied with the changes made by the authors.
- The authors have revised the introduction to reflect the types of models (wrist- vs waist-worn) that were used in the validation studies, and to cite the noted studies. However, we want to underscore that these articles examined agreement of sedentary behavior using accelerometers, as the reviewer notes, and these devices are considered the gold standard for measuring physical activity but not the gold standard for measuring sedentary behavior. Thus, we believe that our manuscript fills an important gap in the literature in that we compare Fitbit measures with those from activPAL devices, which represent the current gold standard approach to measuring sedentary behavior. The revised sentences and their line numbers are included below.
- Please provide the participant characteristics here (line 63)
- We elected to include participant characteristics in the results section as that is our general approach, and we found no guidance in the guidelines for authors about where the journal preferred to see these data. We will defer to the editor as to whether participant characteristics should be moved the methods section rather than retain it in the results section, if that is your preference.
- In my opinion, this aspect is not crucial and can be left in his current form if accepted from the journal.
- Line 81-83: The authors refer to a series of studies using different Fitbit models, including those devices that are worn on the waist. Please provide references for the investigated model if available or justify why citing studies in which different models were used. This aspect can be crucial and must be discussed.
- Lines 93-97: The authors thank the reviewer for calling the discrepancy in models to our attention. The sentence has been changed to “Validation studies have shown wrist worn Fitbit models, such as the Charge 3, to accurately assess the steps of adults in both lab based[18–20] and free living[15,21]”
- I am satisfied with the changes made by the authors.
- Line 124: Please put a reference here for the line “we used the 15-second epoch approach and summed every 4 epochs into minute-by-minute data” (Basset et all 2013)
- The authors realized that “approach” was a misworded description of what was conducted. Instead, “approach” is reworded to “datafile” to better describe the process that was conducted. We were unable to find the reference suggested by the reviewer and were not following a specific data aggregation approach; rather we summed the data into a full minute to compare with the Fitbit minute.
- I misunderstood authors in the previous version and I apologise for this. Some studies used counts·15sec (instead of one minute) to define PA and SB cutpoints. Thus, I am satisfied with the changes made by the authors.
- Line 126-134: Considering the lack of guidelines on the best approach for defining a given minute as sedentary, the authors compared two different methods for classifying sedentary behavior. This is an original aspect investigated in the present study and should be presented in the introduction as an objective of the study and not in the methods section.
- The authors appreciate the reviewer for recognizing this original aspect of the study and the contributions it makes to the field. We have highlighted this component of the manuscript by including the following sentence in the introduction: (line 65-73) In addition, the study examines different approaches to classifying sedentary behavior using activPAL epoch data. The activPAL software provides both event-based and epoch (time-based) data, but there are no guidelines that establish how much of a minute must be spent engaged in sedentary behavior to classify that minute as sedentary. Therefore, we explored two different thresholds (half minute and full minute) to determine which classification scheme agrees most closely with the events data. This comparison advances conversations about how sedentary behavior might best be classified using this gold-standard device and also provided two epoch-based metrics from the activPAL with which to compare Fitbit minute-by-minute data.
- I am satisfied with the changes made by the authors.
- Line 172: Statistical analysis is confusing. Paired sample t-test seems inadequate to compare the variables investigated because there are more than 2 averages.
- Lines 205-208: The authors apologize for the confusion in our description of the statistical methods. We have clarified the sentence to say: Paired t-tests were used to compare the average daily hours of sedentary behavior and percent of the waking day spent sedentary between the Fitbit estimates and the three activPAL estimates (half, full and event), as well as between the pairs of the activPAL estimates. We hope that this clarifies the analysis for the reviewer.
- I found statistical analysis inadequate despite the changes made by the authors. To avoid committing a Type-I error with the average daily hours of sedentary behavior and percent of the waking day spent sedentary between the Fitbit estimates and the three activPAL estimates (half, full, and event), as well as between the pairs of the activPAL estimates, repeated measures one-way analysis of variance (ANOVA) should be used to examine differences in daily hours of SB and percent of the waking day spent sedentary, comparing Fitbit gold standard using three different metrics (half, full and event). Significant overall ANOVA effects should be followed by pairwise comparisons using Bonferroni adjustment.
- Line 175: Did the authors run the Bland Altman test for investigating the agreement? Why did the authors choose ICC? This part is not clear also because the authors use the terms agreement and correspondence interchangeably. When is an agreement considered discordant and when concordant?
- The authors selected ICC because this approach replicates the methods used by other investigators conducting similar research, although we are aware that other investigators have used Bland Altman plots that also serve similar purposes. We elected to use the ICC because it reflects both degrees of correlation and agreement between measures and accommodates the nature of our data in which continuous measures of sedentary time were clustered by days. We are amenable, however, to adding the Bland Altman analyses if the editor believes this would contribute substantial value to the manuscript. We expect that the conclusions will be similar (if not identical) between the two methods. In addition, to help clarify the results, we changed correspondence to agreement throughout the manuscript and altered the sentence below to remove agreement.
- Lines 233-237: Furthermore, classification of individuals as sedentary was the same using the activPAL_Events and the Fitbit for 10 out of the 11 participants, with 9 categorized as sedentary and 1 categorized as non-sedentary (Table 1).
- Results are still confusing. First, the authors stated that compliance was similar (line 220). However, 23.7 + 0.7 vs 20.2 + 1.9 seems quite different, though both high. The authors should run a statistical comparison. Secondly, activPAL_Epoch waking time was 15.9 + 0.8 hours (line 223). Is this independent from the metric adopted (full and half)? Thirdly, if a purpose of the present study is to determine which classification scheme agrees most closely with the events data (line 70), the agreement with ActivPAL_events classification as sedentary should be provided also for activPAL_half and _full. For example, it seems that participant number four is classified as not sedentary only using activPAL_full.
- The authors selected ICC because this approach replicates the methods used by other investigators conducting similar research, although we are aware that other investigators have used Bland Altman plots that also serve similar purposes. We elected to use the ICC because it reflects both degrees of correlation and agreement between measures and accommodates the nature of our data in which continuous measures of sedentary time were clustered by days. We are amenable, however, to adding the Bland Altman analyses if the editor believes this would contribute substantial value to the manuscript. We expect that the conclusions will be similar (if not identical) between the two methods. In addition, to help clarify the results, we changed correspondence to agreement throughout the manuscript and altered the sentence below to remove agreement.
- Table 1 is not well-formatted and I cannot read the right side.
- The authors apologize for the issues with Table 1. It appears that there was a formatting issue. The size of the table has been reduced to ensure it can now fit on the page. If it remains problematic, we can split the information into 2 tables.
- The present table present the agreement between Fitbit and activePAL events only. It should compare also the other metrics.
- Please, when you refer to previous studies, specify the specific model investigated.
- Throughout the paper, the authors have specified whether the Fitbit device used was wrist or waist worn rather than specifying the actual model used in the studies. We used this approach since we believe it provides more purposeful information than the tracker model, since the algorithms used would be similar for all wrist worn devices and all waist worn devices. The types of devices were added in lines 273 and 275 and the sentence now reads “ Two studies suggested that wrist and waist worn Fitbits underestimate sedentary time relative to a non-inclinometer accelerometer [15,20], while another indicated sedentary time captured with a wrist worn Fitbit device is comparable to assessments with a non-inclinometer accelerometer [14].”
- I am satisfied with the changes made by the authors.
